# OMNIPOTENT ADVERSARIAL TRAINING IN THE WILD

## ABSTRACT

Adversarial training is an important topic in robust deep learning, but the community lacks attention to its practical usage. In this paper, we aim to resolve a real-world challenge, i.e., training a model on an imbalanced and noisy dataset to achieve high clean accuracy and adversarial robustness, with our proposed Omnipotent Adversarial Training (`OAT`) strategy. `OAT` consists of two innovative methodologies to address the imperfection in the training set. We first introduce an oracle into the adversarial training process to help the model learn a correct data-label conditional distribution. This carefully-designed oracle can provide correct label annotations for adversarial training. We further propose logits adjustment adversarial training to overcome the data imbalance issue, which can help the model learn a Bayes-optimal distribution. Our comprehensive evaluation results show that `OAT` outperforms other baselines by more than 20% clean accuracy improvement and 10% robust accuracy improvement under complex combinations of data imbalance and label noise scenarios.

## 1 INTRODUCTION

How to enhance the *adversarial robustness* of deep learning models has constantly attracted attention from both industry and academia. Adversarial robustness refers to the ability of a deep learning model to resist against adversarial attacks. Madry et al. (2018) proposed adversarial training (AT), a popular strategy to improve the model's robustness. Due to its high computational cost, numerous works further proposed computation-friendly AT methods (Shafahi et al., 2019; Zheng et al., 2020) which are scalable to large datasets. Although significant efforts have been devoted to making AT more efficient and practical, there still exists a gap for real-world applications. The main obstacle is that these works idealize the training dataset as completely clean and uniformly distributed. However, in reality, annotations are often noisy (Whitehill et al., 2009; Xiao et al., 2015) and datasets tend to be long-tailed (Lin et al., 2017; Wang et al., 2017), making these methods less effective.

Specifically, label noise is a common occurrence in real-world datasets due to variations in the experience and expertise of data annotators. For example, as reported in (Song et al., 2022), the Clothing1M dataset (Xiao et al., 2015) contains about 38.5% noise, and the WebVision dataset (Li et al., 2017) was found to have around 20.0% noise. Although some crowdsourcing platforms, like Amazon Mechanical Turk (MTu, 2022), can provide mechanisms like voting to reduce the ratio of noisy labels in the datasets, it remains challenging to guarantee completely clean label mapping. Consequently, label noise is still an open problem in deep learning model training. On the other hand, data imbalance can occur when it is difficult to collect sufficient samples for several specific classes (Wang et al., 2017). Typically, we call a dataset long-tailed if most of the data belong to several classes, called head classes, and fewer data belong to other classes, known as tail classes (Wang et al., 2017). Given that this is the natural property of the data distribution, it is challenging to create a perfectly balanced dataset in practice. Additionally, label noise can exacerbate data imbalance by introducing additional noise to the tail classes. Thus, it is important to consider both label noise and data imbalance together when developing a robust deep learning model.

Challenges arise when we train a robust model on a noisy and imbalanced dataset. First, in AT, generating adversarial examples (AEs) relies on the gradients, which are calculated with the label and model's prediction, to update the perturbation for the target model. With noisy labels, the generated AEs become less reliable, reducing the effectiveness of AT. Additionally, incorrect annotations prevent the model from learning the correct mapping between data and labels, which harms the clean accuracy of the model. Second, an imbalanced dataset decreases the model's generalizability and makes the model lean to classify a sample into head classes (Lin et al., 2017). This can result in poor performance on tail classes and lower overall robustness of the model.

Most of existing AT solutions only consider clean and balanced datasets. To the best of our knowledge, only two works have examined label noise in the context of AT (Dong et al., 2022b; Huang et al., 2020). However, they aim at addressing the overfitting issue rather than robustness enhancement. The poor label refurbishment effect in these methods under massive label noise makes the models fail to converge during AT (proved in our experiments in Section 4). For the data imbalance scenario, only one published work studies AT on long-tailed datasets (Wu et al., 2021). Since this work pays no attention to the joint effects of label noise and data imbalance on model robustness, it cannot work properly without correct labels, because the label distribution can be misleading.

If we can extract data with wrong annotations in the training set and provide correct labels to them with high probability, we will have the opportunity to mitigate the adverse effects of training models under noisy labels. Furthermore, if we can correct the wrong labels, we will recover a correct label distribution, which is helpful to address the overfitting problem caused by data imbalance. Based on these insights, we propose a novel training strategy, named **O**mnipotent **A**dversarial **T**raining (OAT), which aims to obtain a robust model trained on a noisy and imbalanced dataset. The innovative idea of OAT is to **introduce an *oracle* to regulate the model training over imperfect data samples**.

OAT is a two-step training scheme, i.e., oracle training and robust model training. Specifically, in the first step, we set up an oracle to provide correct annotations for a noisy dataset. Unlike existing label correction methods that rely solely on model predictions (Arazo et al., 2019; Song et al., 2019a), we adopt a novel technique to predict labels using high-dimensional feature embeddings and a $k$-nearest neighbors algorithm. To overcome the data imbalance challenge in oracle training, we propose a dataset re-sampling technique. Moreover, to further improve the label correction process, we adopt the self-supervised contrastive learning technique to train the oracle.

In the second step, to address the data imbalance problem, we introduce the logits adjustment adversarial training, which can help the model learn a Bayes-optimal distribution. By obtaining correct labels from the oracle, we can approximate the true label distribution, which is adopted to adjust the model's predictions, allowing the model to achieve comparable robustness to previous AT methods (Wu et al., 2021). Furthermore, we instruct the model to interact with the oracle to obtain high clean accuracy and robustness even on an imbalanced dataset with massive label noise. Extensive experimental results show that OAT achieves higher clean accuracy and robustness on the noisy and imbalanced training dataset. Overall, our contributions can be summarized as follows.

- We propose the first AT strategy, OAT, aiming to solve a real-world problem, i.e., adversarial training on a noisy and imbalanced dataset.
- OAT outperforms previous works under various practical scenarios. Specifically, it achieves up to 80.72% clean accuracy and 42.84% robust accuracy on a heavy imbalanced dataset with massive label noise, which is about 50% and 20% higher than SOTA methods.
- Our comprehensive experiments can inspire researchers to propose more approaches to minimize the performance gap between ideal and practical datasets.

## 2 PRELIMINARIES

In the following, we provide the necessary definitions before presenting the proposed method. Due to the paper limitation, we leave the discussions of related works and baseline methods in Appendix A.

For a supervised learning algorithm, we consider a dataset with two basic components, i.e., the set of data and the label mapping. We give a formal definition of a dataset[1] as follows:

**Definition 1** *Suppose a set $\mathcal{S}$ and a mapping $\mathcal{A}$ satisfy $\mathcal{A}(x) \in [C]$, where $x \in \mathcal{S}$. The tuple $(\mathcal{S}, \mathcal{A})$ is called a dataset $\mathcal{D}(\mathcal{S}, \mathcal{A})$. $C$ represents the number of classes. $\mathcal{A}(x)$ is the label of data $x$.*

Clearly, given a set $\mathcal{S}$ with the cardinality $|\mathcal{S}|$ and the number of classes $C$, where $|\mathcal{S}| > C$, there are $C + |\mathcal{S}|! \sum_{i=2}^{C} (\binom{C}{i}\binom{|\mathcal{S}|-1}{i-1}(i)!)$ different mappings, where $|\mathcal{S}|!$ and $(i)!$ are the factorial of $|\mathcal{S}|$ and $i$. We introduce a set $\mathfrak{A}$ to represent all possible label mappings $\mathcal{A}$:

**Definition 2** *Given a set $\mathcal{S}$ and the number of classes $C$, $\mathfrak{A}$ contains all mappings $\mathcal{A}$, satisfying $\mathcal{A}(x) \in [C]$ for $x \in \mathcal{S}$.*

---

[1]We leave the open-set problem (Wang et al., 2018) as future work. In this paper, all data with incorrect labels have correct labels within the label set of the dataset (Han et al., 2018a).

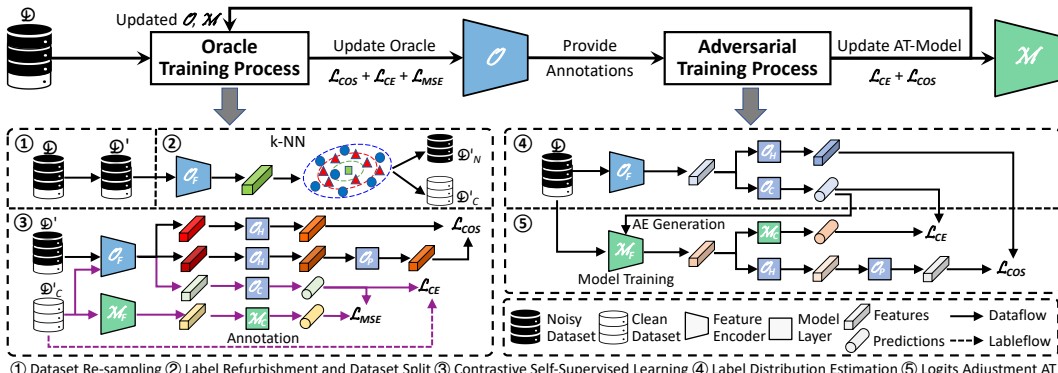

Figure 1: Overview of OAT. We alternately train the oracle and AT-model. The oracle provides the AT-model with new annotations, to overcome the challenges in long-tailed and noisy label learning.

With set $\mathfrak{A}$, we can give a special label mapping $\mathcal{A}_{\mathrm{gt}}$ under certain culture knowledge $\mathfrak{K}$. Every person with knowledge $\mathfrak{K}$ will agree with the output of $\mathcal{A}_{\mathrm{gt}}$ for every $x \in \mathcal{S}$. Then, we call the dataset $\mathcal{D}(\mathcal{S}, \mathcal{A}_{\mathrm{gt}})$ a clean dataset without label noise. Otherwise, any $\mathcal{A} \in \mathfrak{A}$ that is not $\mathcal{A}_{\mathrm{gt}}$ constructs a noisy dataset $\mathcal{D}(\mathcal{S}, \mathcal{A})$. So, whether a dataset contains label noise is depended on $\mathcal{A}$ and independent of $\mathcal{S}$. Formally, we can define the noise ratio (NR) of a dataset $\mathcal{D}(\mathcal{S}, \mathcal{A})$ as $\mathrm{NR} = \frac{\sum_{x \in \mathcal{S}} \mathbb{1}(\mathcal{A}(x) != \mathcal{A}_{\mathrm{gt}}(x))}{|\mathcal{S}|}$, where $|\mathcal{S}|$ is the number of the data in set $\mathcal{S}$. With previous definitions, we can give a formal definition of label distribution for a given dataset $\mathcal{D}(\mathcal{S}, \mathcal{A})$.

**Definition 3** *Given a dataset $\mathcal{D}(\mathcal{S}, \mathcal{A})$, $N_i = \sum_{x \in \mathcal{S}} \mathbb{1}(\mathcal{A}(x) = i)$ representing the number of data in the set $\mathcal{S}$ mapped into class $i$ by $\mathcal{A}$.*

In Definition 3, we count the number of data for each class $i$ based on the output of $\mathcal{A}$. So, given a dataset $\mathcal{D}(\mathcal{S}, \mathcal{A})$, we can calculate its imbalanced ratio (IR) under $\mathcal{A}$: $\mathrm{IR} = \frac{\min(N_i)}{\max(N_i)}$, and the true imbalanced ratio ($\mathrm{IR}_{\mathrm{gt}}$) under $\mathcal{A}_{\mathrm{gt}}$. Usually, if $\mathcal{A} \neq \mathcal{A}_{\mathrm{gt}}$, the label distributions will be different for the clean dataset and noisy datasets. We use $\mathcal{D}$ to represent a dataset if there is no ambiguity.

In practice, obtaining the mapping $\mathcal{A}_{\mathrm{gt}}$ requires lots of additional effort, so the dataset owner usually adopts a plausible mapping $\mathcal{A}$ to approximate the correct mapping, which will introduce label noise into the dataset. Under this situation, both the mapping $\mathcal{A}_{\mathrm{gt}}$ and the corresponding correct label distribution are unknown. So, for AE generation and loss backpropagation in AT, we require reconstructing a more precise label mapping $\mathcal{A}'$ from the known one $\mathcal{A}$ to decrease the label noise in the dataset and calculating the correct label distribution.

## 3 Omnipotent Adversarial Training

To address the label noise and imbalanced data distribution problems, we introduce an oracle $\mathcal{O}$ into the training process to improve the robustness of the AT-model $\mathcal{M}$. This idea is realized with a new training framework, named Omnipotent Adversarial Training (OAT). Figure 1 illustrates the overall workflow of OAT, which consists of two key processes: oracle training (OT) and adversarial training (AT). OAT aims to leverage the oracle $\mathcal{O}$ to provide correct annotations to train an AT-model $\mathcal{M}$ on the dataset $\mathcal{D}$. The oracle can be represented as $\mathcal{O}(\cdot) = \mathcal{O}_C(\mathcal{O}_F(\cdot))$, where $\mathcal{O}_F$ is the feature encoder, and $\mathcal{O}_C$ is the classification layer. The AT-model can be represented as $\mathcal{M}(\cdot) = \mathcal{M}_C(\mathcal{M}_F(\cdot))$, where $\mathcal{M}_F$ is the feature encoder, and $\mathcal{M}_C$ is the classification layer. We use the same architecture for $\mathcal{O}$ and $\mathcal{M}$. In every training epoch, we first train the oracle, then adopt it to predict the labels for the dataset $\mathcal{D}$, and finally use the predictions as annotations to generate AEs and train the AT-model $\mathcal{M}$. Below, we present the details of the OT and AT processes.

### 3.1 Oracle Training

Unlike the traditional model training process that focuses on achieving strong generalizability on test data, oracle training aims to optimize the oracle's ability to predict training samples as accurately as the ground-truth set $\mathcal{A}_{\mathrm{gt}}$. This unique objective motivates us to develop an effective approach to training the oracle. If the oracle is trained under the annotations from the label mapping $\mathcal{A}$,

the training set $\mathcal{D}$ can be both noisy and imbalanced, hindering the oracle's ability to approximate the target mapping $\mathcal{A}_{\text{gt}}$. To address these issues, we introduce four main techniques, i.e., dataset re-sampling, label refurbishment, dataset split, and contrastive self-supervised learning.

**Dataset Re-sampling** (①️ in Figure 1). Training a model to fit an imbalanced label distribution is more challenging than training a model on a balanced one (Lin et al., 2017). Based on this prior, we over-sample the dataset $\mathcal{D}(\mathcal{S}, \mathcal{A})$ to make the number of data for every class equal. Specifically, we first find out the largest number of data among all classes $N_{\max} = \max(N_i)$. For each class $i$, we fix all data $x$, satisfying $\mathcal{A}(x) = i$. So, there will be $N_i$ data in class $i$. Then, we randomly and repeatedly select $N_{\max} - N_i$ data from the fixed data with replacement and add them into the set $\mathcal{S}$ for class $i$. This process yields $N_{\max}$ samples for every class, and we refer to the resulting balanced dataset as $\mathcal{D}'(\mathcal{S}', \mathcal{A})$. The dataset re-sampling process is only launched for the firs time the OT process runs, and the generated set $\mathcal{S}'$ will be used for the following procedure.

**Label Refurbishment and Dataset Split** (②️ in Figure 1). This technique is introduced to improve the prediction accuracy of the oracle $\mathcal{O}$. It has been found that the model first learns samples with correct labels from the noisy dataset (Arpit et al., 2017; Song et al., 2019b). So, in the early training phase, the model gives higher confidence scores for correctly labeled data. Due to the model's generalizability, the samples with incorrect labels will be classified into correct classes with high confidence. Our idea is to use a threshold $\theta_r$ to refurbish labels as follows:

$$\mathcal{A}_r(x) = \begin{cases} \mathcal{A}(x), & \max(\sigma(\mathcal{O}(x))) < \theta_r \\ \arg\max(\sigma(\mathcal{O}(x))), & \max(\sigma(\mathcal{O}(x))) \geq \theta_r \end{cases}$$

where $\mathcal{O}(x)$ is the logits output of data $x$ and $\sigma(\cdot)$ is the softmax function. After label refurbishment, we obtain a dataset $\mathcal{D}'(\mathcal{S}', \mathcal{A}_r)$, which contain less label noise.

To train our oracle as meticulously as possible, we split the dataset $\mathcal{D}'(\mathcal{S}', \mathcal{A}_r)$ into a clean one and a noisy one. Previous works adopt the loss function values (Arazo et al., 2019; Li et al., 2020) or predicted confidence scores (Malach & Shalev-Shwartz, 2017; Song et al., 2019a) to judge whether the data have correct annotations or not, which is not stable and can fail under massive label noise (Feng et al., 2022). Different from them, we adopt a non-parametric $k$-nearest neighbors ($k$-NN) model $\mathcal{K}$ to split the dataset. The insight behind our technique is that models trained in a contrastive self-supervised manner will automatically map the data belonging to the same class into the neighbor feature embedding (Kang et al., 2021), which indicates that data in the same class will have more similar features than data from different classes. Therefore, we first adopt $\mathcal{K}$ to find the $k$-nearest neighbors for each data $x$ in the feature space. Then, we calculate the predicted label $L_x^{\mathcal{K}}$ from $\mathcal{K}$ by finding the class which contains most of the neighbors for each data $x$. If the label $L_x^{\mathcal{K}}$ is the same as $\mathcal{A}_r(x)$, we add $x$ into the clean set $\mathcal{S}_C'$. Otherwise, we add $x$ into the noisy set $\mathcal{S}_N'$. After the label refurbishment and dataset split, we have two new datasets: $\mathcal{D}'(\mathcal{S}_C', \mathcal{A}_r)$ containing less label noise and $\mathcal{D}'(\mathcal{S}_N', \mathcal{A}_r)$ containing more label noise, which are named $\mathcal{D}_C'$ and $\mathcal{D}_N'$, respectively.

**Contrastive Self-Supervised Learning** (③️ in Figure 1). In prior works, models trained in a self-supervised manner are proved to be more robust against label noise (Feng et al., 2022; Karim et al., 2022; Li et al., 2022) and data imbalance (Kang et al., 2021). So, we borrow a contrastive learning approach, BYOL (Grill et al., 2020), but removing the momentum encoder, for two reasons. First, Chen & He (2021) proved that using a shared feature encoder to replace the momentum encoder can also achieve good results. Second, using a shared encoder can improve the efficiency and reduce the training cost. We introduce additional two modules $\mathcal{O}_H$ and $\mathcal{O}_P$ to participate in the contrastive learning part. Because the contrastive learning does not require the labels, we directly adopt the full dataset $\mathcal{D}'$ to train the oracle, and the loss can be represented as:

$$\mathcal{L}_{\text{COS}} = -\mathbb{E}_{x \sim \mathcal{D}'} \frac{\mathcal{O}_H(\mathcal{O}_F(\tau_1(x))) * \mathcal{O}_P(\mathcal{O}_H(\mathcal{O}_F(\tau_2(x))))}{\|\mathcal{O}_H(\mathcal{O}_F(\tau_1(x)))\|_2 * \|\mathcal{O}_P(\mathcal{O}_H(\mathcal{O}_F(\tau_2(x))))\|_2},$$

where $\tau_1$ is a weak data augmentation strategy (only cropping and flipping) and $\tau_2$ is a strong data augmentation strategy based on the AutoAugment (Cubuk et al., 2019).

For the supervised learning part, we only adopt the samples in the previously separated clean dataset $\mathcal{D}_C'$, and the loss is:

$$\mathcal{L}_{\text{CE}} = \mathbb{E}_{x, \mathcal{A}_r(x) \sim \mathcal{D}_C'} \text{cross-entropy}(\mathcal{O}(x), \mathcal{A}_r(x)).$$

Furthermore, to better leverage the knowledge from the oracle, we expect that it can provide the AT-model $\mathcal{M}$ more different prediction distributions from $\mathcal{M}$. So, we adopt a penalty term as follows:

$$\mathcal{L}_{\mathrm{MSE}} = -\mathbb{E}_{x \backsim \mathcal{D}'_C} \mathrm{MSE}(\sigma(\mathcal{O}(x)), \sigma(\mathcal{M}(x)))$$

Overall, the loss function for the oracle training is $\mathcal{L}_{\mathcal{O}} = \mathcal{L}_{\mathrm{COS}} + \mathcal{L}_{\mathrm{CE}} + \mathcal{L}_{\mathrm{MSE}}$.

## 3.2 ADVERSARIAL TRAINING

Although we adopt an oracle to correct the wrong annotations, it is not enough to train a robust model on a dataset with unknown label distributions. Based on a previous study (Wu et al., 2021), it is important to design specific approaches to addressing the dataset imbalance, because the model trained over the long-tailed dataset can badly overfit the head classes. In the AT stage of OAT, we combine two techniques, i.e., label distribution estimation and logits adjustment AT, to address the challenges together.

**Label Distribution Estimation** (④ in Figure 1). As the considered training set can be both noisy and imbalanced, it is important to infer the correct label annotations and label distribution. To obtain a relatively precise label distribution, we first ask the oracle $\mathcal{O}$ to predict the label for each sample in $\mathcal{D}$. To make it clear, we define a new label mapping based on the oracle as follows:

$$\mathcal{A}^{\mathcal{O}}(x) = \arg \max(\sigma(\mathcal{O}(x))), x \in \mathcal{S}.$$

So, the label distribution predicted by the oracle is $N_i^{\mathcal{O}} = \sum_{x \in \mathcal{S}} \mathbb{1}(\mathcal{A}^{\mathcal{O}}(x) = i), i \in [C]$, where $C$ is the number of classes in the dataset $\mathcal{D}$.

**Logits Adjustment AT** (⑤ in Figure 1). To overcome the over-confidence issue in long-tailed recognition, we apply the previous logits adjustment approach (Menon et al., 2021) with the label distribution $N_i^{\mathcal{O}}$. Specifically, we adjust $\mathcal{M}$'s output logits during training in the following way:

$$l = \mathcal{M}(x) + \log([N_1^{\mathcal{O}}, N_2^{\mathcal{O}}, \ldots, N_C^{\mathcal{O}}]).$$

Whether the label distribution is a uniform one or a long-tailed one, the logits adjustment translates the model's confidence scores into Bayes-optimal predictions (Menon et al., 2021) under the current label distribution, making it a universal solution for all possible label distributions.

The logits adjustment AT can be divided into two steps, i.e., AE generation and model training. In the AE generation step, we simply follow PGD-AT (Madry et al., 2018) to generate AEs. This step can be formulated as $x_{\mathrm{adv}} = \mathrm{PGD}(\mathcal{M}, x, \mathcal{A}^{\mathcal{O}}(x))$, where the PGD attack accepts as input a classifier model $\mathcal{M}$, a clean sample $x$ and its corresponding label $\mathcal{A}^{\mathcal{O}}(x)$, and returns an AE $x_{\mathrm{adv}}$. We adjust the output logits during the AE generation.

In the model training step, we consider the oracle as a soft label generator, and adopt its confidence scores as labels to train the AT-model $\mathcal{M}$. It can be seen as a strong and adaptive label smoothing method (Müller et al., 2019), which further addresses the robust overfitting issue (Rice et al., 2020). The loss function is written as

$$\mathcal{L}_{\mathrm{CE}} = -\mathbb{E}_{x \backsim \mathcal{D}} \sum_{i=1}^{C} \log(\sigma(\mathcal{M}(x_{\mathrm{adv}}) + \log([N_1^{\mathcal{O}}, N_2^{\mathcal{O}}, \ldots, N_C^{\mathcal{O}}]))_i) * \sigma(\mathcal{O}(x))_i.$$

To further leverage the feature embedding generated by the oracle, we add a contrastive learning loss into the model training step. This loss has the same formula as the contrastive loss in the oracle training process:

$$\mathcal{L}_{\mathrm{COS}} = -\mathbb{E}_{x \backsim \mathcal{D}} \frac{\mathcal{O}_H(\mathcal{O}_F(x)) * \mathcal{O}_P(\mathcal{O}_H(\mathcal{M}_F(x_{\mathrm{adv}})))}{\|\mathcal{O}_H(\mathcal{O}_F(x))\|_2 * \|\mathcal{O}_P(\mathcal{O}_H(\mathcal{M}_F(x_{\mathrm{adv}})))\|_2},$$

where we consider the PGD attack as a very strong data augmentation strategy.

Overall, the loss function for adversarial training is $\mathcal{L}_{\mathcal{M}} = \mathcal{L}_{\mathrm{CE}} + \mathcal{L}_{\mathrm{COS}}$. In our experiment, we consider $\mathcal{L}_{\mathrm{MSE}}$ in $\mathcal{L}_{\mathcal{O}}$ and $\mathcal{L}_{\mathrm{COS}}$ in $\mathcal{L}_{\mathcal{M}}$ are two terms under the oracle-model interactions. We explore their effectiveness through ablation studies in Section 4.2.

## 4 EXPERIMENTS

### 4.1 CONFIGURATIONS

**Datasets and models.** We adopt two datasets to evaluate `OAT`, i.e., CIFAR-10 and CIFAR-100 (Krizhevsky et al., 2009). We generate imbalanced datasets based on the *exponential method* (Cao et al., 2019), which is widely used in previous papers (Cui et al., 2019; Ren et al., 2020; Wu et al., 2021). For the label noise generation, we consider two types of label noise, i.e., *symmetric noise* and *asymmetric noise*, which are common settings in previous works (Feng et al., 2022; Li et al., 2020; Karim et al., 2022). Specifically, symmetric noise means the noisy label is uniformly selected from all possible labels except the ground-truth one. Asymmetric noise simulates a more practical scenario, where the ground-truth label can only be changed into a new one with similar semantic information, e.g., truck → automobile, bird → airplane, deer → horse, and cat → dog. We only apply the asymmetric noise to CIFAR-10, as we cannot find prior works studying the asymmetric noise in CIFAR-100. When generating a label-noisy and imbalanced dataset, we first build a dataset under the given NR and then use the exponential method on the noisy labels to sample it to obtain a long-tailed dataset under this IR, which can guarantee that all classes contain at least one correct sample. In some cases, the ground-truth label distribution can be a balanced one and the noisy label distribution is badly imbalanced, which increases the difficulty of adversarial training. For the model structure, as the oracle and AT-model in `OAT` are based on ResNet-18 (He et al., 2016), to make a fair comparison, we implement all baseline methods on ResNet-18.

**Baselines.** We consider five baseline methods, i.e., PGD-AT (Madry et al., 2018), TRADES (Zhang et al., 2019), SAT (Huang et al., 2020), TE (Dong et al., 2022b) and RoBal (Wu et al., 2021). Specifically, PGD-AT and TRADES are two representative AT strategies, which are proposed to improve the model's robustness on balanced and clean datasets. SAT and TE study the memorization of AT under random labels. Some of their experimental results are obtained from datasets with random noise and achieve good performance. So we consider that they can be adopted to train models on noisy datasets. In order to make a fair comparison, we adopt the PGD version of SAT and TE, based on their official implementations. RoBal is proposed to solve the long-tailed AT challenge. We compare `OAT` with these baseline methods under various settings.

**Implementation details.** For `OAT`, we adopt the same $k$-NN structure as SSR+ (Feng et al., 2022) with $k = 200$, and follow the hyperparameter setup in its implementation, i.e., $\theta_r = 0.8$. $\mathcal{O}_H$ and $\mathcal{O}_P$ are two MLPs with one hidden layer, whose hidden dimension is 256 and output dimension is 128. We discuss the training cost overhead in Section 4.6.

To evaluate the robustness and clean accuracy of baselines and `OAT`, we follow the training strategy proposed in (Rice et al., 2020), except for RoBal, which follows a different training setting for long-tailed datasets (Wu et al., 2021). All other hyperparameters in baseline methods are set following their official implementations. Specifically, for all methods, we use SGD as the optimizer, with the initial learning rate 0.1, momentum 0.9, weight decay 0.0005, and batch size 128. For RoBal, the total number of training epochs is 80, and we decay the learning rate at the 60-th and 75-th epoch with a factor 0.1. For others, the total number of training epochs is 200, and the learning rate decays at the 100-th and 150-th epoch with a factor 0.1. Note that the learning rate decay is only for the AT-model in `OAT`, while the oracle does not need to adjust the learning rate, because we observe a larger learning rate can slow down the convergence speed of the oracle and improve the AT-model's robustness by introducing uncertainty in the oracle's predictions. For adversarial training, except for TRADES, we adopt $l_\infty$-norm PGD (Madry et al., 2018), with a maximum perturbation size $\epsilon = 8/255$ for 10 iterations, and step length $\alpha = 2/255$ in each iteration. For TRADES, we follow its official implementation, with a maximum perturbation size $\epsilon = 8/255$ for 10 iterations, step length $\alpha = 2/255$ in each iteration, and robust loss scale $\beta = 6.0$.

**Metrics.** We mainly report the clean accuracy (CA) and robust accuracy (RA) under AutoAttack (Croce & Hein, 2020). The results under other different attacks can be found in Appendix D. We save the "**Best**" model with the highest robustness on the test set under PGD-20 and the "**Last**" model at the end of training. Due to page limit, some results of the "**Last**" models are in Appendix B.

### 4.2 ABLATION STUDY

We first explore the effectiveness of different components proposed in `OAT`, including the oracle-model interactions and logits adjustment. Table 1 presents the results on a balanced and imbalanced

| Method | Best | | Last | | Best | | Last | |
|---|---|---|---|---|---|---|---|---|
| | CA | RA | CA | RA | CA | RA | CA | RA |
| Setup | IR = 1.0; NR = 0.0 | | | | IR = 0.02; NR = 0.0 | | | |
| $\mathcal{O} + \mathcal{M}$ | 82.73 | 48.74 | 85.01 | 46.11 | - | - | - | - |
| w/ interaction | 83.15 | 48.80 | 85.10 | 47.44 | 63.10 | 29.96 | 63.95 | 27.68 |
| w/ logits adjustment | 83.49 | 48.49 | 85.44 | 47.25 | 74.46 | 31.33 | 68.70 | 24.60 |

Table 1: Ablation of components in OAT on CIFAR-10.

| Noise Type = symmetric | NR = 0.0 | | | | NR = 0.2 | | | | NR = 0.4 | | | | NR = 0.6 | | | | NR = 0.8 | | | |
|---|---|---|---|---|---|---|---|---|---|---|---|---|---|---|---|---|---|---|---|---|
| | Best | | Last | | Best | | Last | | Best | | Last | | Best | | Last | | Best | | Last | |
| | CA | RA | CA | RA | CA | RA | CA | RA | CA | RA | CA | RA | CA | RA | CA | RA | CA | RA | CA | RA |
| PGD-AT | 82.92 | 47.83 | 84.44 | 41.90 | 79.90 | 46.83 | 78.10 | 32.78 | 74.80 | 44.88 | 73.09 | 32.44 | 66.97 | 40.70 | 64.21 | 32.65 | - | - | - | - |
| TRADES | 82.88 | 48.63 | 82.84 | 46.45 | 79.89 | 45.56 | 78.40 | 40.76 | 76.95 | 42.40 | 73.47 | 31.72 | 72.66 | 37.62 | 64.54 | 18.06 | - | - | - | - |
| SAT | 72.79 | 45.39 | 70.61 | 44.37 | 69.82 | 44.54 | 67.89 | 43.18 | 65.50 | 43.26 | 63.21 | 40.34 | 50.91 | 36.30 | 47.43 | 31.79 | - | - | - | - |
| TE | 82.49 | 50.37 | 83.00 | 49.33 | 80.71 | 49.12 | 81.09 | 47.42 | 77.32 | 46.80 | 77.57 | 44.75 | 65.51 | 42.50 | 66.45 | 38.90 | - | - | - | - |
| RoBal | 81.73 | 46.92 | 84.58 | 46.54 | 76.18 | 45.90 | 80.23 | 45.31 | 70.66 | 43.89 | 74.70 | 43.50 | 51.88 | 36.17 | 51.63 | 35.95 | - | - | - | - |
| OAT | **83.49** | 48.49 | **85.44** | 47.25 | **83.99** | 48.13 | **85.16** | 47.05 | **83.69** | **48.58** | **85.40** | **47.57** | **83.00** | **48.57** | **84.81** | **46.91** | **82.24** | **48.14** | **84.44** | **46.91** |

Table 2: Results on balanced CIFAR-10 with asymmetric label noise. The best results are in **bold**. "-" means the model does not converge under this setting.

| Noise Type = asymmetric | NR = 0.2 | | NR = 0.4 | | NR = 0.6 | |
|---|---|---|---|---|---|---|
| | CA | RA | CA | RA | CA | RA |
| PGD-AT | 80.84 | 46.99 | 76.22 | 45.59 | 51.83 | 35.01 |
| TRADES | 78.83 | 45.96 | 69.14 | 39.99 | 50.37 | 34.29 |
| SAT | 67.88 | 43.77 | 59.25 | 38.88 | 52.41 | 34.94 |
| TE | 79.41 | **49.39** | 71.59 | 43.52 | 51.69 | 35.57 |
| RoBal | 80.78 | 45.58 | 77.74 | 45.19 | 70.73 | 39.97 |
| OAT | **83.47** | 48.56 | **83.65** | **48.82** | **71.99** | **43.06** |

Table 3: Results from "**Best**" models on balanced CIFAR-10 with the asymmetric label noise.

| Noise Type = symmetric | NR = 0.0 | | NR = 0.2 | | NR = 0.4 | | NR = 0.6 | | NR = 0.8 | |
|---|---|---|---|---|---|---|---|---|---|---|
| | CA | RA | CA | RA | CA | RA | CA | RA | CA | RA |
| PGD-AT | 57.01 | 24.76 | 51.81 | 23.08 | 46.28 | 21.44 | 33.83 | 17.86 | - | - |
| TRADES | 56.65 | 22.75 | 52.82 | 20.40 | 48.00 | 17.30 | 42.22 | 14.18 | - | - |
| SAT | 41.37 | 21.29 | 38.77 | 20.44 | 34.46 | 18.74 | 26.68 | 15.48 | - | - |
| TE | 57.06 | 24.91 | 51.66 | 23.43 | 46.21 | 21.43 | 33.86 | 18.01 | - | - |
| RoBal | 56.17 | 24.18 | 51.37 | 23.22 | 45.10 | 20.76 | 34.79 | 17.39 | - | - |
| OAT | **59.14** | **25.79** | **58.75** | **25.72** | **57.82** | **25.72** | **56.95** | **25.01** | **53.89** | **24.73** |

Table 4: Results from "**Best**" models on balanced CIFAR-100 with asymmetric label noise.

clean dataset, respectively. It is clear that with the oracle-model interaction, both clean accuracy and robust accuracy are improved. Furthermore, the results indicate that the interaction can mitigate robust overfitting. On the other hand, the logits adjustment will harm the clean accuracy and robustness of models trained on the balanced dataset and cause some robust overfitting on the imbalanced dataset, because the estimated label distribution from the oracle is not as exact as the ground-truth distribution. However, when we train models on an imbalanced dataset, the clean accuracy and robustness of the best model indicate that the effectiveness of the logits adjustment is significant. Overall, both oracle-model interaction and logits adjustment are essential components.

## 4.3 RESULTS UNDER LABEL NOISE

We evaluate the models trained on balanced but noisy datasets. Tables 2 and 3 show the results of the balanced CIFAR-10 dataset containing symmetric and asymmetric noise, respectively. Table 4 illustrates the results of models trained on the balanced CIFAR-100 dataset with symmetric noise. Symmetric noise can harm the clean accuracy of baseline models to a bigger degree than harming the robustness. Clearly, decreasing the clean accuracy will reduce the robust accuracy. So when the noise ratio reaches 0.8, we observe models trained with baseline methods do not converge, and the robustness is close to zero. Based on the results, it is clear that OAT achieves consistent high clean accuracy and robust accuracy under different settings. Specifically, SAT adopts the model's confidence scores to refurbish the labels, and achieves lower clean accuracy, as the model trained with AEs will be less overconfident of the data (Wen et al., 2020) and have slower convergence speed, making the label refurbishment fail. On the other hand, TE only works under less label noise and fails when there are massive noise in the dataset. For example, on CIFAR-10 and NR = 0.6, the clean accuracy of the model with the best robust accuracy of OAT is about 32% higher than that of SAT. The robustness of this model is about 6% higher than that of TE. Besides, with the increasing noise

| NR = 0.0 | CIFAR-10 | | | | | | CIFAR-100 | | | | | |
|---|---|---|---|---|---|---|---|---|---|---|---|---|
| | IR = 0.1 | | IR = 0.05 | | IR = 0.02 | | IR = 0.1 | | IR = 0.05 | | IR = 0.02 | |
| | CA | RA | CA | RA | CA | RA | CA | RA | CA | RA | CA | RA |
| PGD-AT | 72.27 | 35.31 | 65.88 | 31.79 | - | - | 42.59 | 14.85 | 38.47 | 12.89 | - | - |
| TRADES | 64.46 | 34.65 | 55.84 | 30.63 | - | - | 39.41 | 16.23 | 34.38 | 14.03 | - | - |
| SAT | 66.32 | 34.95 | 56.31 | 29.99 | - | - | 34.42 | 17.60 | 30.63 | 15.56 | - | - |
| TE | 67.38 | 35.93 | 57.58 | 32.16 | - | - | 42.58 | 14.83 | 38.14 | 12.94 | - | - |
| RoBal | 75.93 | 38.54 | 71.71 | 36.71 | 65.89 | **32.01** | 43.43 | 16.94 | 39.19 | 14.59 | 34.31 | 12.18 |
| OAT | **79.42** | **41.69** | **75.82** | **38.15** | **74.46** | 31.33 | **50.10** | **19.10** | **46.88** | **16.66** | **41.82** | **14.18** |

Table 5: Results from "**Best**" models on clean but imbalanced CIFAR-10 and CIFAR-100.

| Noise Type = | CA | RA | CA | RA | CA | RA | CA | RA | CA | RA | CA | RA |
|---|---|---|---|---|---|---|---|---|---|---|---|---|
| *symmetric* | IR=0.1 | | IR=0.1 | | IR=0.05 | | IR=0.05 | | IR=0.02 | | IR=0.02 | |
| | NR=0.4 | | NR=0.6 | | NR=0.4 | | NR=0.6 | | NR=0.4 | | NR=0.6 | |
| PGD-AT | 48.97 | 28.87 | 31.42 | 20.97 | 36.58 | 24.60 | - | - | - | - | - | - |
| TRADES | 44.44 | 23.91 | 30.93 | 20.22 | 33.06 | 21.65 | - | - | - | - | - | - |
| SAT | 37.99 | 26.94 | 18.69 | 16.70 | 28.12 | 22.71 | - | - | - | - | - | - |
| TE | 45.04 | 28.56 | 20.62 | 17.10 | 33.78 | 24.11 | - | - | - | - | - | - |
| RoBal | 55.13 | 37.00 | 32.14 | 25.20 | 52.25 | 34.17 | 28.96 | 22.61 | 47.29 | 30.04 | 28.06 | 22.01 |
| OAT | **80.07** | **42.86** | **80.72** | **42.84** | **79.07** | **41.25** | **79.10** | **40.64** | **76.13** | **37.48** | **73.54** | **35.60** |

Table 6: Results from "**Best**" models on imbalanced and noisy CIFAR-10 (symmetric).

| Noise Type = | CA | RA | CA | RA | CA | RA | CA | RA | CA | RA | CA | RA |
|---|---|---|---|---|---|---|---|---|---|---|---|---|
| *symmetric* | IR=0.1 | | IR=0.1 | | IR=0.05 | | IR=0.05 | | IR=0.02 | | IR=0.02 | |
| | NR=0.4 | | NR=0.6 | | NR=0.4 | | NR=0.6 | | NR=0.4 | | NR=0.6 | |
| PGD-AT | 23.24 | 10.26 | 19.98 | 9.38 | 18.59 | 8.95 | 13.53 | 8.02 | - | - | - | - |
| TRADES | 22.27 | 8.67 | 16.95 | 7.21 | 22.27 | 7.30 | 14.42 | 6.20 | - | - | - | - |
| SAT | 25.37 | 13.41 | 17.01 | 10.25 | 21.63 | 11.53 | 14.44 | 9.33 | - | - | - | - |
| TE | 23.40 | 10.05 | 19.68 | 8.97 | 18.53 | 9.04 | 14.14 | 7.90 | - | - | - | - |
| RoBal | 28.83 | 12.50 | 16.59 | 8.52 | 24.35 | 10.61 | 12.29 | 6.29 | 19.25 | 7.87 | 10.58 | 4.20 |
| OAT | **49.99** | **19.86** | **48.50** | **18.83** | **46.53** | **17.06** | **42.79** | **16.20** | **39.77** | **13.71** | **35.68** | **12.67** |

Table 7: Results from "**Best**" models on imbalanced and noisy CIFAR-100 (symmetric).

ratio, both clean accuracy and robustness face the overfitting challenge. Among all methods, OAT achieves the best results to alleviate overfitting, due to the adaptive label smoothing from the oracle.

## 4.4 RESULTS UNDER DATA IMBALANCE

We then assess the models trained on imbalanced clean datasets. In long-tailed recognition, the main challenge is the overfitting problem, where the model gives high confidence scores to head classes. Table 5 displays the performance of models trained on long-tailed CIFAR-10 and CIFAR-100. In this setting, the training algorithms only need to address the long-tailed challenges. Hence, RoBal, which is specifically designed for long-tailed AT, achieves competitive results compared with OAT. On the other hand, OAT outperforms RoBal in two aspects: *consistency* and *generalization*. First, OAT achieves better clean accuracy and robust accuracy on different datasets and different IR values. For example, on CIFAR-10 and IR = 0.05, the clean and robust accuracy of the "**Best**" model from OAT is about 4% and 1% higher than the ones from RoBal. On CIFAR-100 and IR = 0.02, our "**Best**" model achieves 41.82% clean accuracy and 14.18% robust accuracy, which are 7% and 2% higher than that of RoBal. Second, RoBal requires different hyperparameters for CIFAR-10 and CIFAR-100, but OAT does not require changing the hyperparameters. Overall, for the long-tailed AT task, OAT is more advanced than RoBal.

## 4.5 RESULTS UNDER LABEL NOISE AND DATA IMBALANCE

Finally, we study the models trained on imbalanced and noisy datasets. Tables 6 and 7 present the results on imbalanced datasets containing symmetric noise. Table 8 shows the results on imbalanced CIFAR-10 with asymmetric noise. We consider various combinations of IR (selected from {0.1, 0.05, 0.02}) and NR (selected from {0.4, 0.6}). Results of other setups are in Appendix C and E.

We observe that OAT outperforms other baselines in both clean accuracy and robustness under various setups and datasets. One important reason is that previous methods cannot correctly predict the

| Noise Type = | CA | RA | CA | RA | CA | RA | CA | RA | CA | RA | CA | RA |
|---|---|---|---|---|---|---|---|---|---|---|---|---|
| *asymmetric* | IR=0.1 | | IR=0.1 | | IR=0.05 | | IR=0.05 | | IR=0.02 | | IR=0.02 | |
| | NR=0.4 | | NR=0.6 | | NR=0.4 | | NR=0.6 | | NR=0.4 | | NR=0.6 | |
| PGD-AT | 59.69 | 31.36 | 55.96 | 29.24 | 54.17 | 28.97 | 47.36 | 26.55 | - | - | - | - |
| TRADES | 55.54 | 28.26 | 51.30 | 27.07 | 47.54 | 25.27 | 43.13 | 23.69 | - | - | - | - |
| SAT | 53.88 | 30.75 | 51.01 | 29.39 | 50.78 | 28.15 | 45.90 | 26.32 | - | - | - | - |
| TE | 58.68 | 31.34 | 53.66 | 29.06 | 52.22 | 28.85 | 47.36 | 26.56 | - | - | - | - |
| RoBal | 69.05 | 35.84 | 65.86 | 33.14 | 63.62 | 31.96 | 57.90 | 29.99 | 56.16 | 27.82 | 56.35 | 27.87 |
| OAT | **79.03** | **42.09** | **69.44** | **37.88** | **76.71** | **38.96** | **66.19** | **35.00** | **70.67** | **30.83** | **62.39** | **29.06** |

Table 8: Results from "**Best**" models on imbalanced and noisy CIFAR-10 (asymmetric).

label distribution for an imbalanced and noisy dataset, which hinders the AE generation process. Without valid AEs and corresponding labels to train the model, either clean accuracy or robustness will significantly decrease. In contrast, the oracle in OAT can naturally predict the label distribution because of the four techniques we propose in the oracle training process. As a result, OAT can achieve both higher clean accuracy and robust accuracy. For example, on CIFAR-10, IR = 0.05, NR = 0.6 of symmetric noise, the clean accuracy and robust accuracy of the "**Best**" model from OAT are about 27% and 7% higher than the ones of RoBal, respectively.

Asymmetric noise can transform the dataset from a balanced one into an imbalanced one. For example, under asymmetric noise, the number of samples in class "truck" will be significantly smaller than that in class "automobile". RoBal achieves better results than other baselines. However, because of the label distribution estimation and logits adjustment in OAT, it outperforms RoBal in both clean accuracy and robustness, which proves that OAT is the best choice for different types of label noise.

## 4.6 TRAINING COST OVERHEAD

We compare the training time cost between OAT and PGD-AT on one RTX 3090 GPU card. We implement our code with Pytorch version 1.12, and Cuda version 11.6. When we train a model on CIFAR-10, the training time per epoch is 110 seconds for PGD-AT. For OAT, the oracle training time per epoch is 39 seconds, and the adversarial training time per epoch is 116 seconds. So, the total training time for one epoch is 155 seconds, which is only 45 seconds longer than the PGD-AT. Considering the clean accuracy and robustness we obtain with OAT, the training cost is acceptable.

## 4.7 LABEL DISTRIBUTION CORRECTION

To evaluate the quality of the estimated label distribution, Figure 2 illustrates the oracle's predicted labels in the 10th, 50th, and 100th training epoch, respectively. Here "Prior" denotes the label distribution of the known dataset, and "GT" denotes the ground-truth distribution of clean labels, which is unknown for a noisy dataset. We consider a complex case, where both clean labels and noisy labels are long-tailed. Other cases can be found in Appendix F. The results prove that our oracle can correctly produce the label distribution in this scenario, which facilitates the significant improvement of clean accuracy and robustness.

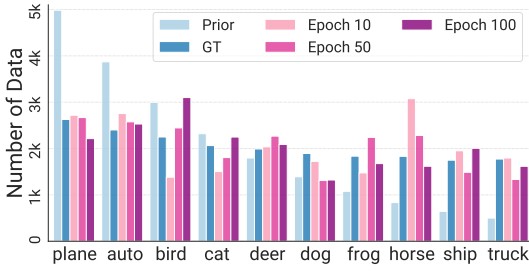

Figure 2: Label distribution predicted on the noisy and imbalanced CIFAR-10 dataset.

## 5 CONCLUSION AND FUTURE WORK

We propose a new training strategy, OAT, to solve real-world adversarial training challenges, including label noise and data imbalance. By introducing an oracle, OAT achieves state-of-the-art results under different evaluation setups. We hope the dataset re-sampling, logits adjustment AT and other proposed techniques can inspire researchers to explore more effective training strategies for practical usage.

The main limitation of OAT is the performance drop under massive asymmetric noise, although it is much better than prior works. From the results, we can find that models trained on a dataset containing massive asymmetric label noise will have lower clean accuracy and become easier to overfit the training set. It is important to address this challenge as future work.

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

## A    RELATED WORKS

**Noisy Label Recognition.** Label noise is a common threat in practice because the data annotation process heavily depends on the knowledge of the workers. Recently, numerous works aim to address the label noise in image recognition from different perspectives, including new model architectures (Sukhbaatar & Fergus, 2015), robust loss functions (Wang et al., 2019; Zhang & Sabuncu, 2018), label correction (Huang et al., 2020; Reed et al., 2015) and sample selection (Han et al., 2018b). Specifically, Goldberger & Ben-Reuven (2017) proposed a noise adaptation layer to model the label transition pattern with a noise transition matrix. However, the estimation error between the adaptation layer and real label noise distribution is large when the noise rate is high in the training set, causing worse results. For the robust loss functions, Ghosh et al. (2017) proved that the Mean Absolute Error (MAE) loss is robust to the label noise, but it harms the model's generalizability. Label correction (Huang et al., 2020; Reed et al., 2015) is another way to address the label noise problem. Existing methods aim to learn the correct label mapping and then correct the wrong labels. Li et al. (2020) proposed a sample selection method, adopting two models to adaptively choose samples with smaller loss values as clean data and samples with larger loss values as noisy data. Then, each model predicts a label for the noisy data and provides them to its peer model to learn together with clean data.

**Long-tailed Recognition.** Data imbalance is common in collected large datasets, since data belonging to some categories are naturally rare, e.g., special diseases in medical datasets (Skin-7 (Codella et al., 2018)), endangered species in animal datasets (iNaturalist 2018 (iNa, 2018)). Such imbalanced data distribution will harm the model's generalizability (Buda et al., 2018). Long-tailed recognition is proposed to solve this real-world problem and train models on imbalanced datasets. A straightforward approach is to re-sample the training distribution to make it more balance, such as random under-sampling head classes (Liu et al., 2009) and random over-sampling tail classes (Han et al., 2005). Recently, a logits adjustment method is proposed (Menon et al., 2021; Ren et al., 2020), solving the dilemma that models lean to classify samples into head classes with high probability.

**Adversarial Training.** Adversarial training (AT) (Madry et al., 2018; Zhang et al., 2019) is one of the most famous approaches to increase the robustness of models. It generates on-the-fly AEs to train the models. Recently, several works are proposed to promote AT in real-world applications. Zheng et al. (2020) proposed an efficient AT method based on the transferability of AEs to reduce the AE generation cost, making it possible to adopt AT on large datasets, such as ImageNet (Deng et al., 2009). Dong et al. (2022a) study the label shifting in adversarial training to address the overfitting problem. However, their work is not related to the topic in this paper, and we do not consider it as a baseline method. Researchers also studied the behaviors of models trained on randomly labeled datasets with AT and found that models trained with AT can memorize those random labels (Dong et al., 2022b; Huang et al., 2020). Based on the observation, they proposed new training algorithms to address the overfitting problem, which can also be adopted to train models on noisy datasets. For another practical problem, RoBal (Wu et al., 2021) is proposed to meet the imbalanced dataset scenario. **To the best of our knowledge, there is no work focusing on training models on both imbalanced and noisy datasets with AT. We step forward to real-world applications and explore this threat model in this paper.** Our method combines label refurbishment and distribution re-balancing, achieving state-of-the-art results under different combinations of label noise and data imbalance settings.

## B    FULL TABLES OF MAIN PAPER

Due to the page limit, we cannot show the whole tables in our main paper. So, we give the full results in this supplementary materials for readers' further reference. These tables contain more results under different configurations, and the results prove the advantages of OAT in both clean accuracy and robustness. Specifically, we show the full results of models trained on balanced but noisy datasets in Tables 9, and 10. The results in Tables 11 and 12 are for models trained on clean but imbalanced datasets. In Tables 13, 14, and 15, the models are trained on imbalanced and noisy datasets for further evaluation of the complex scenarios.

| Noise Type = *symmetric* | NR = 0.0 | | | | NR = 0.2 | | | | NR = 0.4 | | | | NR = 0.6 | | | | NR = 0.8 | | | |
|---|---|---|---|---|---|---|---|---|---|---|---|---|---|---|---|---|---|---|---|---|
| | Best | | Last | | Best | | Last | | Best | | Last | | Best | | Last | | Best | | Last | |
| Method | CA | RA | CA | RA | CA | RA | CA | RA | CA | RA | CA | RA | CA | RA | CA | RA | CA | RA | CA | RA |
| PGD-AT | 57.01 | 24.76 | 57.03 | 19.27 | 51.81 | 23.08 | 46.65 | 12.93 | 46.28 | 21.44 | 35.90 | 7.32 | 33.83 | 17.86 | 22.98 | 3.42 | - | - | - | - |
| TRADES | 56.65 | 22.75 | 54.44 | 22.13 | 52.82 | 20.40 | 48.29 | 17.15 | 48.00 | 17.30 | 40.16 | 11.63 | 42.22 | 14.18 | 28.35 | 5.44 | - | - | - | - |
| SAT | 41.37 | 21.29 | 36.99 | 20.00 | 38.77 | 20.44 | 34.30 | 18.93 | 34.46 | 18.74 | 28.74 | 17.32 | 26.68 | 15.48 | 18.17 | 12.00 | - | - | - | - |
| TE | 57.06 | 24.91 | 57.05 | 20.34 | 51.66 | 23.43 | 47.56 | 14.32 | 46.21 | 21.43 | 37.65 | 9.10 | 33.86 | 18.01 | 24.41 | 4.47 | - | - | - | - |
| RoBal | 56.17 | 24.18 | 58.29 | 22.98 | 51.37 | 23.22 | 52.49 | 20.30 | 45.10 | 20.76 | 45.81 | 17.94 | 34.79 | 15.19 | 34.68 | 17.30 | - | - | - | - |
| OAT | 59.14 | 25.79 | 58.89 | 24.69 | 58.75 | 25.72 | 58.51 | 24.40 | 57.82 | 25.72 | 57.88 | 24.65 | 56.95 | 25.01 | 56.80 | 24.63 | 53.89 | 24.73 | 54.49 | 23.88 |

Table 9: Results on balanced CIFAR-100 dataset, in which the label noise is symmetric.

| Noise Type = *asymmetric* | NR = 0.2 | | | | NR = 0.4 | | | | NR = 0.6 | | | |
|---|---|---|---|---|---|---|---|---|---|---|---|---|
| | Best | | Last | | Best | | Last | | Best | | Last | |
| Method | CA | RA | CA | RA | CA | RA | CA | RA | CA | RA | CA | RA |
| PGD-AT | 80.84 | 46.99 | 80.56 | 39.91 | 76.22 | 45.59 | 75.84 | 38.27 | 51.83 | 35.01 | 53.23 | 31.54 |
| TRADES | 78.83 | 45.96 | 78.94 | 42.85 | 69.14 | 39.99 | 67.84 | 36.37 | 50.37 | 34.29 | 53.64 | 33.77 |
| SAT | 67.88 | 43.77 | 64.22 | 42.25 | 59.25 | 38.88 | 51.06 | 37.61 | 52.41 | 34.94 | 47.35 | 33.81 |
| TE | 79.41 | 49.39 | 80.17 | 47.75 | 71.59 | 43.52 | 64.32 | 40.51 | 51.69 | 35.57 | 50.70 | 34.37 |
| RoBal | 80.78 | 45.58 | 82.70 | 45.22 | 77.74 | 45.19 | 80.37 | 44.30 | 70.73 | 39.97 | 72.11 | 40.03 |
| OAT | 83.47 | 48.56 | 84.85 | 46.61 | 83.65 | 48.82 | 85.03 | 47.14 | 71.99 | 43.06 | 73.94 | 42.36 |

Table 10: Results on balanced CIFAR-10 dataset, in which the label noise is asymmetric.

| Method | IR = 1.0 | | | | IR = 0.1 | | | | IR = 0.05 | | | | IR = 0.02 | | | |
|---|---|---|---|---|---|---|---|---|---|---|---|---|---|---|---|---|
| | Best | | Last | | Best | | Last | | Best | | Last | | Best | | Last | |
| | CA | RA | CA | RA | CA | RA | CA | RA | CA | RA | CA | RA | CA | RA | CA | RA |
| PGD-AT | 82.92 | 47.83 | 84.44 | 41.90 | 72.27 | 35.31 | 73.91 | 29.70 | 65.88 | 31.79 | 67.18 | 26.81 | - | - | - | - |
| TRADES | 82.88 | 48.63 | 82.84 | 46.45 | 64.46 | 34.65 | 69.88 | 32.30 | 55.84 | 30.63 | 62.26 | 28.62 | - | - | - | - |
| SAT | 72.79 | 45.39 | 70.61 | 44.37 | 66.32 | 34.95 | 51.06 | 31.94 | 56.31 | 29.99 | 43.12 | 28.46 | - | - | - | - |
| TE | 82.49 | 50.37 | 83.00 | 49.33 | 67.38 | 35.93 | 67.29 | 34.85 | 57.58 | 32.16 | 57.73 | 30.97 | - | - | - | - |
| RoBal | 81.73 | 46.92 | 84.58 | 46.54 | 75.93 | 38.54 | 77.80 | 36.70 | 71.71 | 36.71 | 73.64 | 32.78 | 65.89 | 32.01 | 68.41 | 29.17 |
| OAT | 83.49 | 48.49 | 85.44 | 47.25 | 79.42 | 41.69 | 79.96 | 36.76 | 75.82 | 38.15 | 77.83 | 32.71 | 74.46 | 31.33 | 68.70 | 24.60 |

Table 11: Results on clean but imbalanced CIFAR-10 dataset.

| Method | IR = 1.0 | | | | IR = 0.1 | | | | IR = 0.05 | | | | IR = 0.02 | | | |
|---|---|---|---|---|---|---|---|---|---|---|---|---|---|---|---|---|
| | Best | | Last | | Best | | Last | | Best | | Last | | Best | | Last | |
| | CA | RA | CA | RA | CA | RA | CA | RA | CA | RA | CA | RA | CA | RA | CA | RA |
| PGD-AT | 57.01 | 24.76 | 57.03 | 19.27 | 42.59 | 14.85 | 42.78 | 13.06 | 38.47 | 12.89 | 37.94 | 11.97 | - | - | - | - |
| TRADES | 56.65 | 22.75 | 54.44 | 22.13 | 39.41 | 16.23 | 40.46 | 14.47 | 34.38 | 14.03 | 36.20 | 13.09 | - | - | - | - |
| SAT | 41.37 | 21.29 | 36.99 | 20.00 | 34.42 | 17.60 | 31.80 | 16.63 | 30.63 | 15.56 | 28.53 | 14.85 | - | - | - | - |
| TE | 57.06 | 24.91 | 57.05 | 20.34 | 42.58 | 14.83 | 41.83 | 13.26 | 38.14 | 12.94 | 37.97 | 11.83 | - | - | - | - |
| RoBal | 56.17 | 24.18 | 58.29 | 22.98 | 43.43 | 16.94 | 44.34 | 14.99 | 39.19 | 14.59 | 40.70 | 13.58 | 34.31 | 12.18 | 36.32 | 11.53 |
| OAT | 59.14 | 25.79 | 58.89 | 24.69 | 50.10 | 19.10 | 49.93 | 18.42 | 46.88 | 16.66 | 46.30 | 16.02 | 41.82 | 14.18 | 41.27 | 14.05 |

Table 12: Results on clean but imbalanced CIFAR-100 dataset.

| Noise Type = *symmetric* | Best | | Last | | Best | | Last | | Best | | Last | | Best | | Last | | Best | | Last | | Best | | Last | |
|---|---|---|---|---|---|---|---|---|---|---|---|---|---|---|---|---|---|---|---|---|---|---|---|---|
| | CA | RA | CA | RA | CA | RA | CA | RA | CA | RA | CA | RA | CA | RA | CA | RA | CA | RA | CA | RA | CA | RA | CA | RA |
| Method | IR = 0.1; NR = 0.4 | | | | IR = 0.1; NR = 0.6 | | | | IR = 0.05; NR = 0.4 | | | | IR = 0.05; NR = 0.6 | | | | IR = 0.02; NR = 0.4 | | | | IR = 0.02; NR = 0.6 | | | |
| PGD-AT | 48.97 | 28.87 | 46.57 | 13.28 | 31.42 | 20.97 | 30.38 | 17.02 | 36.58 | 24.60 | 37.42 | 13.74 | - | - | - | - | - | - | - | - | - | - | - | - |
| TRADES | 44.44 | 23.91 | 46.00 | 16.62 | 30.93 | 20.22 | 32.42 | 11.70 | 33.06 | 21.65 | 38.13 | 16.33 | - | - | - | - | - | - | - | - | - | - | - | - |
| SAT | 37.99 | 26.94 | 27.32 | 21.77 | 18.69 | 16.70 | 15.08 | 12.71 | 28.12 | 22.71 | 22.94 | 19.28 | - | - | - | - | - | - | - | - | - | - | - | - |
| TE | 45.04 | 28.56 | 42.25 | 25.67 | 20.62 | 17.10 | 20.75 | 16.98 | 33.78 | 24.11 | 32.40 | 22.14 | - | - | - | - | - | - | - | - | - | - | - | - |
| RoBal | 55.13 | 37.00 | 60.20 | 35.88 | 32.14 | 25.20 | 32.14 | 25.20 | 52.25 | 34.17 | 54.64 | 33.29 | 28.96 | 22.61 | 27.70 | 21.23 | 47.29 | 30.04 | 48.56 | 29.39 | 28.06 | 22.01 | 26.96 | 21.19 |
| OAT | 80.07 | 42.86 | 80.24 | 39.16 | 80.72 | 42.84 | 81.04 | 39.66 | 79.07 | 41.25 | 79.28 | 36.64 | 79.10 | 40.64 | 79.14 | 37.17 | 76.13 | 37.48 | 75.89 | 32.65 | 73.54 | 35.60 | 71.67 | 30.16 |

Table 13: Results on imbalanced and noisy CIFAR-10 dataset, in which the label noise is symmetric.

| Noise Type = *symmetric* | Best | | Last | | Best | | Last | | Best | | Last | | Best | | Last | | Best | | Last | | Best | | Last | |
|---|---|---|---|---|---|---|---|---|---|---|---|---|---|---|---|---|---|---|---|---|---|---|---|---|
| | CA | RA | CA | RA | CA | RA | CA | RA | CA | RA | CA | RA | CA | RA | CA | RA | CA | RA | CA | RA | CA | RA | CA | RA |
| Method | IR = 0.1; NR = 0.4 | | | | IR = 0.1; NR = 0.6 | | | | IR = 0.05; NR = 0.4 | | | | IR = 0.05; NR = 0.6 | | | | IR = 0.02; NR = 0.4 | | | | IR = 0.02; NR = 0.6 | | | |
| PGD-AT | 23.24 | 10.26 | 23.55 | 5.14 | 19.98 | 9.38 | 13.95 | 2.45 | 18.59 | 8.95 | 21.16 | 4.49 | 13.53 | 8.02 | 12.58 | 2.07 | - | - | - | - | - | - | - | - |
| TRADES | 22.27 | 8.67 | 25.63 | 6.37 | 16.95 | 7.21 | 16.88 | 3.36 | 22.27 | 7.30 | 22.12 | 5.70 | 14.42 | 6.20 | 15.01 | 2.99 | - | - | - | - | - | - | - | - |
| SAT | 25.37 | 13.41 | 22.99 | 12.70 | 17.01 | 10.25 | 14.00 | 9.45 | 21.63 | 11.53 | 19.64 | 11.06 | 14.44 | 9.33 | 12.95 | 8.42 | - | - | - | - | - | - | - | - |
| TE | 23.40 | 10.05 | 24.23 | 5.15 | 19.68 | 8.97 | 13.97 | 2.69 | 18.53 | 9.04 | 21.50 | 4.83 | 14.14 | 7.90 | 12.62 | 2.24 | - | - | - | - | - | - | - | - |
| RoBal | 28.83 | 12.50 | 29.72 | 9.02 | 16.59 | 8.52 | 18.01 | 7.66 | 24.35 | 10.61 | 25.85 | 8.82 | 12.29 | 6.29 | 13.53 | 5.79 | 19.25 | 7.87 | 21.74 | 6.93 | 10.58 | 4.20 | 10.61 | 3.78 |
| OAT | 49.99 | 19.86 | 49.38 | 18.64 | 48.50 | 18.83 | 46.96 | 18.44 | 46.53 | 17.06 | 45.40 | 16.57 | 42.79 | 16.20 | 42.16 | 15.45 | 39.77 | 13.71 | 39.60 | 13.61 | 35.68 | 12.67 | 35.62 | 12.27 |

Table 14: Results on imbalanced and noisy CIFAR-100 dataset, in which the label noise is symmetric.

# C  OTHER SETUPS FOR IMBALANCED AND NOISY DATASETS

Besides the settings discussed in our main paper, i.e., the IR is selected from {0.1, 0.05, 0.02} and the NR is selected from {0.4, 0.6}, we show the results of NR is 0.2 under different IRs. The results in Tables 16, 17, and 18 are for CIFAR-10 with symmetric noise, CIFAR-10 with asymmetric noise, and CIFAR-100 with symmetric noise, respectively. The results prove that OAT outperforms all baselines under various setups.

| Noise Type = *asymmetric* | Best | | Last | | Best | | Last | | Best | | Last | | Best | | Last | | Best | | Last | | Best | | Last | |
|---|---|---|---|---|---|---|---|---|---|---|---|---|---|---|---|---|---|---|---|---|---|---|---|---|
| | CA | RA | CA | RA | CA | RA | CA | RA | CA | RA | CA | RA | CA | RA | CA | RA | CA | RA | CA | RA | CA | RA | CA | RA |
| Method | IR = 0.1; NR = 0.4 | | | | IR = 0.1; NR = 0.6 | | | | IR = 0.05; NR = 0.4 | | | | IR = 0.05; NR = 0.6 | | | | IR = 0.02; NR = 0.4 | | | | IR = 0.02; NR = 0.6 | | | |
| PGD-AT | 59.69 | 31.36 | 60.40 | 23.35 | 55.96 | 29.24 | 54.96 | 22.09 | 54.17 | 28.97 | 55.09 | 21.69 | 47.36 | 26.55 | 50.33 | 20.15 | - | - | - | - | - | - | - | - |
| TRADES | 55.54 | 28.26 | 58.54 | 25.57 | 51.30 | 27.07 | 52.42 | 24.22 | 47.54 | 25.27 | 51.48 | 23.02 | 43.13 | 23.69 | 47.23 | 21.59 | - | - | - | - | - | - | - | - |
| SAT | 53.88 | 30.75 | 44.08 | 27.96 | 51.01 | 29.39 | 39.13 | 25.32 | 50.78 | 28.15 | 47.46 | 25.15 | 45.90 | 26.32 | 30.37 | 21.80 | - | - | - | - | - | - | - | - |
| TE | 58.68 | 31.34 | 56.20 | 30.33 | 53.66 | 29.06 | 51.21 | 27.66 | 52.22 | 28.85 | 48.90 | 27.68 | 47.36 | 26.56 | 43.20 | 24.26 | - | - | - | - | - | - | - | - |
| RoBal | 69.05 | 35.84 | 71.23 | 32.52 | 65.86 | 33.14 | 64.68 | 29.63 | 63.62 | 31.96 | 65.71 | 28.78 | 57.90 | 29.99 | 61.06 | 27.63 | 56.16 | 27.82 | 59.28 | 26.01 | 56.35 | 27.87 | 56.28 | **24.97** |
| OAT | **79.03** | **42.09** | **80.01** | **38.21** | **69.44** | **37.88** | **69.68** | **33.57** | **76.71** | **38.96** | **76.98** | **33.82** | **66.19** | **35.00** | **66.77** | **29.83** | **70.67** | **30.83** | **70.35** | **26.60** | **62.39** | **29.06** | **61.52** | 21.42 |

Table 15: Results on imbalanced and noisy CIFAR-10 dataset, in which the label noise is asymmetric.

| Noise Type = *symmetric* | Best | | Last | | Best | | Last | | Best | | Last | |
|---|---|---|---|---|---|---|---|---|---|---|---|---|
| | CA | RA | CA | RA | CA | RA | CA | RA | CA | RA | CA | RA |
| Method | IR = 0.1; NR = 0.2 | | | | IR = 0.05; NR = 0.2 | | | | IR = 0.02; NR = 0.2 | | | |
| PGD-AT | 63.39 | 32.35 | 60.80 | 18.78 | 53.75 | 28.91 | 51.74 | 18.13 | - | - | - | - |
| TRADES | 54.06 | 27.91 | 58.24 | 23.79 | 46.22 | 25.31 | 49.95 | 21.72 | - | - | - | - |
| SAT | 54.65 | 30.36 | 40.46 | 27.25 | 43.67 | 27.15 | 33.30 | 24.33 | - | - | - | - |
| TE | 61.61 | 32.40 | 57.29 | 30.14 | 51.42 | 28.73 | 45.86 | 26.86 | - | - | - | - |
| RoBal | 66.79 | 38.93 | 70.70 | 36.47 | 62.04 | 36.04 | 66.80 | 33.44 | 56.15 | 31.93 | 60.24 | 29.87 |
| OAT | **79.57** | **42.69** | **80.58** | **38.57** | **77.93** | **39.75** | **78.82** | **36.39** | **74.03** | **36.09** | **76.13** | **31.99** |

Table 16: Results on imbalanced and noisy CIFAR-10 dataset, in which the label noise is symmetric.

| Noise Type = *asymmetric* | Best | | Last | | Best | | Last | | Best | | Last | |
|---|---|---|---|---|---|---|---|---|---|---|---|---|
| | CA | RA | CA | RA | CA | RA | CA | RA | CA | RA | CA | RA |
| Method | IR = 0.1; NR = 0.2 | | | | IR = 0.05; NR = 0.2 | | | | IR = 0.02; NR = 0.2 | | | |
| PGD-AT | 64.49 | 32.24 | 64.12 | 24.64 | 55.98 | 29.30 | 58.18 | 22.59 | - | - | - | - |
| TRADES | 58.24 | 30.77 | 60.33 | 27.82 | 50.43 | 27.47 | 54.78 | 24.75 | - | - | - | - |
| SAT | 58.03 | 31.70 | 46.11 | 29.00 | 53.15 | 29.27 | 38.88 | 26.39 | - | - | - | - |
| TE | 58.81 | 32.81 | 58.31 | 31.37 | 54.50 | 29.43 | 51.05 | 28.51 | - | - | - | - |
| RoBal | 72.88 | 37.02 | 74.04 | 35.07 | 67.63 | 35.05 | 70.99 | 31.96 | 62.10 | 31.09 | 64.95 | 28.58 |
| OAT | **79.50** | **41.87** | **80.39** | **37.88** | **75.56** | **38.66** | **77.70** | **34.40** | **73.32** | **33.49** | **73.28** | **29.52** |

Table 17: Results on imbalanced and noisy CIFAR-10 dataset, in which the label noise is asymmetric.

| Noise Type = *symmetric* | Best | | Last | | Best | | Last | | Best | | Last | |
|---|---|---|---|---|---|---|---|---|---|---|---|---|
| | CA | RA | CA | RA | CA | RA | CA | RA | CA | RA | CA | RA |
| Method | IR = 0.1; NR = 0.2 | | | | IR = 0.05; NR = 0.2 | | | | IR = 0.02; NR = 0.2 | | | |
| PGD-AT | 28.52 | 12.17 | 33.09 | 8.77 | 22.69 | 10.26 | 29.52 | 8.02 | - | - | - | - |
| TRADES | 33.26 | 12.35 | 32.87 | 10.53 | 28.92 | 11.45 | 28.71 | 9.14 | - | - | - | - |
| SAT | 30.10 | 15.79 | 27.97 | 15.04 | 26.93 | 13.95 | 25.14 | 13.39 | - | - | - | - |
| TE | 28.52 | 12.18 | 32.72 | 8.82 | 22.83 | 10.06 | 29.24 | 7.71 | - | - | - | - |
| RoBal | 37.72 | 15.04 | 37.37 | 12.22 | 32.84 | 12.88 | 33.61 | 10.76 | 28.21 | 10.62 | 28.86 | 8.97 |
| OAT | **50.34** | **19.23** | **50.36** | **18.72** | **46.50** | **17.10** | **46.58** | **16.59** | **40.78** | **14.32** | **40.48** | **13.95** |

Table 18: Results on imbalanced and noisy CIFAR-100 dataset, in which the label noise is symmetric.

## D    OTHER ATTACKS

Besides AutoAttack (Croce & Hein, 2020), we consider other $L_\infty$-norm and $L_2$-norm attacks to evaluate the robustness of the models trained with OAT. Specifically, in Tables 19, 21, and 23, we show the results of models under four $L_\infty$-norm attacks, i.e., PGD-20, PGD-100 (Madry et al., 2018), CW-100 (Carlini & Wagner, 2017) and AutoAttack (AA) (Croce & Hein, 2020). For CW attacks, we replace the CE loss in PGD attacks with CW loss. The attack settings are $\epsilon = 8/255$ and $\eta = 2/255$. The number of attack steps is 20 for PGD-20, and 100 for PGD-100 and CW-100. In Tables 20, 22, and 24, we show the results of models under three $L_2$-norm attacks. For the PGD attacks, the max perturbation size is $\epsilon = 0.5$, and the step length is $\alpha = 0.1$. We consider the 20-step attack, PGD-20, and the 100-step attack, PGD-100. For the CW attack, we replace the CE loss in PGD attack with CW loss. Overall, under both $L_\infty$-norm and $L_2$-norm attacks, the models trained with OAT achieving high clean accuracy and robust accuracy, which proves that OAT is an advanced strategy for addressing the data imbalance and label noise challenges in adversarial training.

| Method | CA | RA | | | | CA | RA | | | | CA | RA | | | |
|---|---|---|---|---|---|---|---|---|---|---|---|---|---|---|---|
| | | PGD-20 | PGD-100 | CW-100 | AA | | PGD-20 | PGD-100 | CW-100 | AA | | PGD-20 | PGD-100 | CW-100 | AA |
| OAT | IR = 1.0; NR = 0.0 | | | | | IR = 1.0; NR = 0.2 | | | | | IR = 1.0; NR = 0.4 | | | | |
| | 83.49 | 52.73 | 52.33 | 50.36 | 48.49 | 83.99 | 52.31 | 51.97 | 50.41 | 48.13 | 83.69 | 52.72 | 52.31 | 50.57 | 48.58 |
| OAT | IR = 0.1; NR = 0.0 | | | | | IR = 0.1; NR = 0.2 | | | | | IR = 0.1; NR = 0.4 | | | | |
| | 79.42 | 45.15 | 44.94 | 43.37 | 41.69 | 79.57 | 46.04 | 45.60 | 44.40 | 42.69 | 80.07 | 46.77 | 46.57 | 44.77 | 42.86 |
| OAT | IR = 0.05; NR = 0.0 | | | | | IR = 0.05; NR = 0.2 | | | | | IR = 0.05; NR = 0.4 | | | | |
| | 75.82 | 42.22 | 42.02 | 39.86 | 38.15 | 77.93 | 43.67 | 43.42 | 41.60 | 39.75 | 79.07 | 44.50 | 44.19 | 43.02 | 41.25 |
| OAT | IR = 0.02; NR = 0.0 | | | | | IR = 0.02; NR = 0.2 | | | | | IR = 0.02; NR = 0.4 | | | | |
| | 74.46 | 35.50 | 35.11 | 32.83 | 31.33 | 74.03 | 40.29 | 40.05 | 37.83 | 36.09 | 76.13 | 40.97 | 40.59 | 39.45 | 37.48 |

Table 19: Results under $L_\infty$ attacks on CIFAR-10 dataset, in which the label noise is symmetric. Results are from "**Best**" models.

| Method | CA | RA | | | CA | RA | | | CA | RA | | |
|---|---|---|---|---|---|---|---|---|---|---|---|---|
| | | PGD-20 | PGD-100 | CW-100 | | PGD-20 | PGD-100 | CW-100 | | PGD-20 | PGD-100 | CW-100 |
| OAT | IR = 1.0; NR = 0.0 | | | | IR = 1.0; NR = 0.2 | | | | IR = 1.0; NR = 0.4 | | | |
| | 83.49 | 64.13 | 63.43 | 61.81 | 83.99 | 64.10 | 63.56 | 61.45 | 83.69 | 63.60 | 62.99 | 61.28 |
| OAT | IR = 0.1; NR = 0.0 | | | | IR = 0.1; NR = 0.2 | | | | IR = 0.1; NR = 0.4 | | | |
| | 79.42 | 58.39 | 57.89 | 56.12 | 79.57 | 57.99 | 57.65 | 56.45 | 80.07 | 59.65 | 59.21 | 57.63 |
| OAT | IR = 0.05; NR = 0.0 | | | | IR = 0.05; NR = 0.2 | | | | IR = 0.05; NR = 0.4 | | | |
| | 75.82 | 54.90 | 54.64 | 52.80 | 77.93 | 57.15 | 56.74 | 55.08 | 79.07 | 57.10 | 56.63 | 55.32 |
| OAT | IR = 0.02; NR = 0.0 | | | | IR = 0.02; NR = 0.2 | | | | IR = 0.02; NR = 0.4 | | | |
| | 74.46 | 51.08 | 50.72 | 48.69 | 74.03 | 53.78 | 53.55 | 51.69 | 76.13 | 54.77 | 54.38 | 53.11 |

Table 20: Results under $L_2$ attacks on CIFAR-10 dataset, in which the label noise is symmetric. Results are from "**Best**" models.

| Method | CA | RA | | | | CA | RA | | | | CA | RA | | | |
|---|---|---|---|---|---|---|---|---|---|---|---|---|---|---|---|
| | | PGD-20 | PGD-100 | CW-100 | AA | | PGD-20 | PGD-100 | CW-100 | AA | | PGD-20 | PGD-100 | CW-100 | AA |
| OAT | IR = 1.0; NR = 0.0 | | | | | IR = 1.0; NR = 0.2 | | | | | IR = 1.0; NR = 0.4 | | | | |
| | 83.49 | 52.73 | 52.33 | 50.36 | 48.49 | 83.47 | 52.66 | 52.31 | 50.50 | 48.56 | 83.65 | 52.84 | 52.48 | 51.04 | 48.82 |
| OAT | IR = 0.1; NR = 0.0 | | | | | IR = 0.1; NR = 0.2 | | | | | IR = 0.1; NR = 0.4 | | | | |
| | 79.42 | 45.15 | 44.94 | 43.37 | 41.69 | 79.50 | 45.60 | 45.08 | 43.79 | 41.87 | 79.03 | 45.83 | 45.56 | 43.88 | 42.09 |
| OAT | IR = 0.05; NR = 0.0 | | | | | IR = 0.05; NR = 0.2 | | | | | IR = 0.05; NR = 0.4 | | | | |
| | 75.82 | 42.22 | 42.02 | 39.86 | 38.15 | 75.56 | 42.36 | 42.06 | 40.64 | 38.66 | 76.71 | 42.63 | 42.40 | 41.00 | 38.96 |
| OAT | IR = 0.02; NR = 0.0 | | | | | IR = 0.02; NR = 0.2 | | | | | IR = 0.02; NR = 0.4 | | | | |
| | 74.46 | 35.50 | 35.11 | 32.83 | 31.33 | 73.32 | 37.66 | 37.31 | 35.37 | 33.49 | 70.67 | 34.92 | 34.61 | 33.07 | 30.83 |

Table 21: Results under $L_\infty$ attacks on CIFAR-10 dataset, in which the label noise is asymmetric. Results are from "**Best**" models.

| Method | CA | RA | | | CA | RA | | | CA | RA | | |
|---|---|---|---|---|---|---|---|---|---|---|---|---|
| | | PGD-20 | PGD-100 | CW-100 | | PGD-20 | PGD-100 | CW-100 | | PGD-20 | PGD-100 | CW-100 |
| OAT | IR = 1.0; NR = 0.0 | | | | IR = 1.0; NR = 0.2 | | | | IR = 1.0; NR = 0.4 | | | |
| | 83.49 | 64.13 | 63.43 | 61.81 | 83.47 | 63.61 | 63.10 | 61.23 | 83.65 | 63.95 | 63.45 | 61.64 |
| OAT | IR = 0.1; NR = 0.0 | | | | IR = 0.1; NR = 0.2 | | | | IR = 0.1; NR = 0.4 | | | |
| | 79.42 | 58.39 | 57.89 | 56.12 | 79.50 | 58.19 | 57.80 | 56.49 | 79.03 | 59.06 | 58.59 | 56.92 |
| OAT | IR = 0.05; NR = 0.0 | | | | IR = 0.05; NR = 0.2 | | | | IR = 0.05; NR = 0.4 | | | |
| | 75.82 | 54.90 | 54.64 | 52.80 | 75.56 | 55.30 | 55.11 | 53.33 | 76.71 | 56.34 | 56.09 | 54.45 |
| OAT | IR = 0.02; NR = 0.0 | | | | IR = 0.02; NR = 0.2 | | | | IR = 0.02; NR = 0.4 | | | |
| | 74.46 | 51.08 | 50.72 | 48.69 | 73.32 | 51.66 | 51.38 | 49.60 | 70.67 | 48.77 | 48.46 | 46.42 |

Table 22: Results under $L_2$ attacks on CIFAR-10 dataset, in which the label noise is asymmetric.

| Method | CA | RA | | | | CA | RA | | | | CA | RA | | | |
|---|---|---|---|---|---|---|---|---|---|---|---|---|---|---|---|
| | | PGD-20 | PGD-100 | CW-100 | AA | | PGD-20 | PGD-100 | CW-100 | AA | | PGD-20 | PGD-100 | CW-100 | AA |
| OAT | IR = 1.0; NR = 0.0 | | | | | IR = 1.0; NR = 0.2 | | | | | IR = 1.0; NR = 0.4 | | | | |
| | 59.14 | 30.37 | 30.20 | 27.80 | 25.79 | 58.75 | 30.05 | 29.84 | 27.67 | 25.72 | 57.82 | 29.92 | 29.67 | 27.57 | 25.72 |
| OAT | IR = 0.1; NR = 0.0 | | | | | IR = 0.1; NR = 0.2 | | | | | IR = 0.1; NR = 0.4 | | | | |
| | 50.10 | 23.45 | 23.40 | 20.59 | 19.10 | 50.34 | 23.60 | 23.42 | 20.78 | 19.23 | 49.99 | 23.70 | 23.63 | 21.29 | 19.86 |
| OAT | IR = 0.05; NR = 0.0 | | | | | IR = 0.05; NR = 0.2 | | | | | IR = 0.05; NR = 0.4 | | | | |
| | 46.88 | 20.60 | 20.50 | 18.18 | 16.66 | 46.50 | 21.05 | 20.92 | 18.73 | 17.10 | 46.53 | 21.14 | 20.98 | 18.44 | 17.06 |
| OAT | IR = 0.02; NR = 0.0 | | | | | IR = 0.02; NR = 0.2 | | | | | IR = 0.02; NR = 0.4 | | | | |
| | 41.82 | 17.60 | 17.53 | 15.35 | 14.18 | 40.78 | 17.45 | 17.34 | 15.27 | 14.32 | 39.77 | 17.39 | 17.39 | 14.82 | 13.71 |

Table 23: Results under $L_\infty$ attacks on CIFAR-10 dataset, in which the label noise is symmetric. Results are from "**Best**" models.

# E    OAT UNDER EXTREME SETTINGS

Besides the experimental setups discussed in our main paper, we further consider more challenging and extreme label noise and data imbalance configurations. In Table 25, we consider that the 80% labels in datasets are incorrect. The results prove that OAT can still achieve high clean accuracy and robustness under various data imbalance ratios, while other baseline methods cannot converge under such massive label noise.

| Method | CA | RA | | | CA | RA | | | CA | RA | | |
|---|---|---|---|---|---|---|---|---|---|---|---|---|
| | | PGD-20 | PGD-100 | CW-100 | | PGD-20 | PGD-100 | CW-100 | | PGD-20 | PGD-100 | CW-100 |
| OAT | IR = 1.0; NR = 0.0 | | | | IR = 1.0; NR = 0.2 | | | | IR = 1.0; NR = 0.4 | | | |
| | 59.14 | 39.95 | 39.64 | 37.43 | 58.75 | 40.12 | 39.77 | 37.32 | 57.82 | 39.34 | 39.15 | 36.93 |
| OAT | IR = 0.1; NR = 0.0 | | | | IR = 0.1; NR = 0.2 | | | | IR = 0.1; NR = 0.4 | | | |
| | 50.10 | 32.93 | 32.78 | 29.98 | 50.34 | 33.01 | 32.89 | 30.21 | 49.99 | 33.29 | 33.11 | 30.48 |
| OAT | IR = 0.05; NR = 0.0 | | | | IR = 0.05; NR = 0.2 | | | | IR = 0.05; NR = 0.4 | | | |
| | 46.88 | 29.54 | 29.41 | 27.18 | 46.50 | 30.45 | 30.37 | 27.87 | 46.53 | 30.61 | 30.46 | 27.97 |
| OAT | IR = 0.02; NR = 0.0 | | | | IR = 0.02; NR = 0.2 | | | | IR = 0.02; NR = 0.4 | | | |
| | 41.82 | 26.22 | 26.10 | 23.96 | 40.78 | 25.32 | 25.18 | 23.25 | 39.77 | 24.68 | 24.61 | 22.39 |

Table 24: Results under $L_2$ attacks on CIFAR-100 dataset, in which the label noise is symmetric. Results are from "**Best**" models.

| Method | CIFAR-10 | | | | | CIFAR-100 | | | | |
|---|---|---|---|---|---|---|---|---|---|---|
| | CA | RA | | | | CA | RA | | | |
| | | PGD-20 | PGD-100 | CW-100 | AA | | PGD-20 | PGD-100 | CW-100 | AA |
| OAT | IR = 1.0; NR = 0.8 | | | | | IR = 1.0; NR = 0.8 | | | | |
| | 82.24 | 51.98 | 51.82 | 50.03 | 48.14 | 53.89 | 28.60 | 28.45 | 26.42 | 24.73 |
| OAT | IR = 0.1; NR = 0.8 | | | | | IR = 0.1; NR = 0.8 | | | | |
| | 78.18 | 45.98 | 45.69 | 44.21 | 42.26 | 39.78 | 19.73 | 19.75 | 17.61 | 16.62 |
| OAT | IR = 0.05; NR = 0.8 | | | | | IR = 0.05; NR = 0.8 | | | | |
| | 70.51 | 38.40 | 38.16 | 36.37 | 34.47 | 31.45 | 14.43 | 14.32 | 12.46 | 11.64 |
| OAT | IR = 0.02; NR = 0.8 | | | | | IR = 0.02; NR = 0.8 | | | | |
| | 54.68 | 30.16 | 30.16 | 27.48 | 26.56 | 25.56 | 11.15 | 11.13 | 9.49 | 8.97 |

Table 25: Results of OAT under massive (symmetric) label noise settings. All attacks are in $L_\infty$-norm. Results are from "**Best**" models.

## F LABEL DISTRIBUTION CORRECTION

To evaluate the quality of the estimated label distribution, we illustrate the oracle's predicted labels in Figure 3. We use "Prior" to represent the label distribution of the known dataset, and "GT" to represent the ground-truth distribution of clean labels, which is unknown for a noisy dataset. We plot the estimated label distribution in the 10th, the 50th, and the 100th training epoch, respectively. In Figure 3a and Figure 3b, we show the estimated distribution for clean datasets. The results prove that our oracle can correctly predict balanced and imbalanced label distribution. On the other hand, in Figure 3c and Figure 3d, we plot the label distribution of noisy datasets. Specifically, in Figure 3c, the ground-truth labels are almost balanced, and the noisy labels are long-tailed. In Figure 3d, both clean labels and noisy labels are long-tailed. The results prove that our oracle can correctly produce the label distribution under complex scenarios. So, OAT outperforms other baselines in various settings.

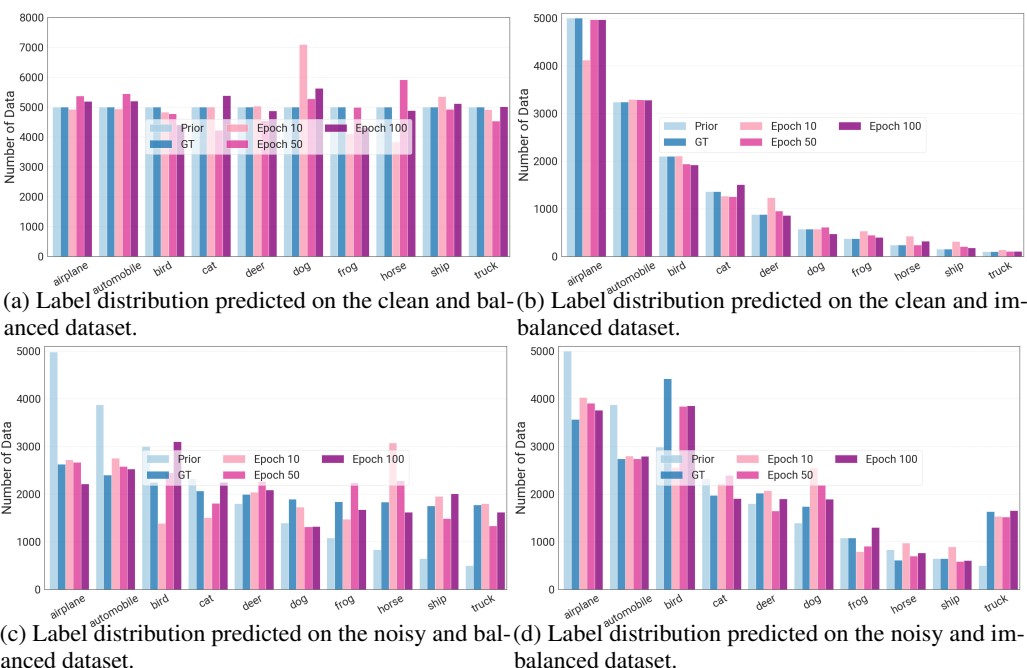

(a) Label distribution predicted on the clean and balanced dataset.

(b) Label distribution predicted on the clean and imbalanced dataset.

(c) Label distribution predicted on the noisy and balanced dataset.

(d) Label distribution predicted on the noisy and imbalanced dataset.

Figure 3: The estimated label distribution in the 10-th, the 50-th, and the 100-th epoch from the oracle.

