# OpenReview forum: "Omnipotent Adversarial Training in the Wild"
_ICLR.cc/2024/Conference — ICLR 2024 Conference Withdrawn Submission_

### Official Review · Reviewer_TYeL · 2023-10-31

**Soundness:** 3 good
**Presentation:** 3 good
**Contribution:** 2 fair
**Rating:** 5
**Confidence:** 4

**Summary:**

This paper tackles the challenges associated with noisy and imbalanced training data in a scenario of adversarial training. The proposed method is composed of an oracle training process (resampling, refurbishment & split, and contrastive learning) and adversarial training (distribution estimation and logit adjustment). Experimental results using CIFAR-10 and 100 shows much better clean accuracy (+20% points) and robust accuracy (+10% points) than the baseline methods.

**Strengths:**

+ This paper is the first trial to achieve better clean and robust accuracy for adversarial training under the noisy and imbalanced conditions.
+ Multiple techniques are efficiently combined for better adversarial training and the key components in this work are all reasonable.
+ The effectiveness of the proposed method is clearly shown with CIFAR-10 and 100 by significantly outperforming the baseline methods in terms of both clean accuracy and robust accuracy.

**Weaknesses:**

1. I am afraid the technical novelty is relatively weak.
- Data-resampling (over-sampling, sub-sampling, SMOTE, etc) can be found in many papers.
- The label refurbishment and dataset split using k-NN seems essentially the same as in the following paper. If not, the authors should clarify the novelty more clearly.
[a] Dara Bahri, Heinrich Jiang, and Maya Gupta. Deep k-nn for noisy labels, Proceedings of Machine Learning Research, 2020.
[b] Chen Feng, Georgios Tzimiropoulos, Ioannis Patras. SSR: An Efficient and Robust Framework for Learning with Unknown Label Noise, BMVC, 2022.
-- Contrastive self-supervised learning is also well-known.
-- Label distribution estimation, i.e., label reassignment, is commonly used as in [b].
-- Logit adjustment is simply taken from [Menon, ICLR21]

2. The experimental validation is weak
- The experiments are conducted only with CIFAR-10 and 100. I admit these two datasets are standard, but the authors might want to show more results with larger datasets to give greater impact to the community.
- The baselines are all simple adversarial training methods. As the authors might be aware, there are already many methods for noisy labels and imbalanced data. The authors might want to at least apply such methods to the baselines. Otherwise, I am afraid that I cannot say the comparison is fair.

**Questions:**

As pointed out in the weakness part, I have concerns about the technical novelty. Please clarify which part was technically novel.


Just as a reference, an adversarial training method for imbalanced datasets has been presented recently. Since this paper was officially published in Oct. 2023, I am NOT requesting the authors to compare their work to this one.

Wentao Wang, Harry Shomer, Yuxuan Wan, Yaxin Li, Jiangtao Huang, and Hui Liu. 2023. A Mix-up Strategy to Enhance Adversarial Training with Imbalanced Data. In Proceedings of the 32nd ACM International Conference on Information and Knowledge Management (CIKM '23).

---

### Official Review · Reviewer_j17w · 2023-11-01

**Soundness:** 2 fair
**Presentation:** 2 fair
**Contribution:** 2 fair
**Rating:** 5
**Confidence:** 4

**Summary:**

This paper addressed the challenges of adversarial training from noisy labeled and class imbalanced dataset. A two stage method was proposed to sequentially update oracle model for label correction and adversarial training on imbalanced data. Comparisons to state-of-the-art adversarial training methods demonstrate the effectiveness of the method.

**Strengths:**

Strength:

1. This paper explores a new setting for adversarial training where training data could be contaminated with noise and imbalanced.

**Weaknesses:**

Weakness:

1. The proposed method does not unify adversarial training with learning from noisy and long-tailed data. These two parts are seemly optimized separately.

2. The techniques introduced for learning from noisy and long-tailed data are not new. Re-sampling is a well-known method for learning from imbalanced data. Label refurbishment is also widely adopted in self-training. Contrastive training is also quite matured. The technical contribution is thus very limited.

3. The evaluation of adversarial attack is not strong enough. PGD is a relatively weak attack method by now. Strong methods, e.g. AutoAttack, should be evaluated additionally.

4. The ablation study is weak. It is hard to see which components are most effective from the presentation in Tab.1.

**Questions:**

It is necessary to elaborate more on the contributions and novelties.

Evaluations against stronger adversarial attacks are necessary.

A clear ablation study is encourage to better analyse effectiveness of proposed method.

---

### Official Review · Reviewer_zhDe · 2023-11-04

**Soundness:** 3 good
**Presentation:** 3 good
**Contribution:** 2 fair
**Rating:** 3
**Confidence:** 4

**Summary:**

This paper aims to develop an adversarial training for a model on an imbalanced and noisy dataset. The proposed method first introduces an oracle to correct the data labels. To overcome the data imbalance challenge, a dataset re-sampling technique is used. To further improve the label correction process, the self-supervised contrastive learning technique is adopted to train the oracle. Then, the adversarial training is carried out.

**Strengths:**

The problems of label noise, sample imbalance, and adversarial sample are simultaneously considered in training a model.

**Weaknesses:**

Sample labeling noise, sample imbalance, and adversarial sample are problems that have been studied separately, and a number of related methods have been proposed. The proposed method in this draft is a simple combination of existing techniques. The sample label correction, data re-sampling, contrastive learning, and adversarial training included in the proposed method are conventional methods. Overall, not much technological innovation has been seen in this manuscript.

**Questions:**

In addition to combining pre-existing technologies to solve multiple problems, what are the original innovations that stem from this manuscript?